



# Impacts of different types of El Niño events on water quality over the Corn Belt, United States

Pan Chen[1,2], Wenhong Li[2,*] and Keqi He[2]

[1] College of Water Resources Science and Engineering, Taiyuan University of Technology, Taiyuan 030024, China
[2] Earth and Climate Sciences, Nicholas School of the Environment, Duke University, NC 27708, USA

*Correspondence to*: Wenhong Li (wenhong.li@duke.edu)

**Abstract.** The United States Corn Belt region, which primarily includes two large basins, namely, the Ohio-Tennessee River Basin (OTRB) and the Upper Mississippi River Basin (UMRB), is responsible for the Gulf of Mexico hypoxic zone. Climate patterns such as El Niño can affect the runoff and thus the water quality over the Corn Belt. In this study, the impacts of eastern Pacific (EP) and central Pacific (CP) El Niños on water quality over the Corn Belt region were analyzed using the Soil and Water Assessment Tool (SWAT) models. Our results indicated that at the outlets, annual total nitrogen (TN) and total phosphorus (TP) loads decreased by 13.1% and 14.0% at OTRB, 18.5% and 19.8% at UMRB, respectively, during the EP-El Niño years, whereas during the CP-El Niño years, they increased by 3.3% and 4.6% at OTRB, 5.7% and 4.4% at UMRB, respectively. On the sub-basin scales, more sub-basins showed negative (positive) anomalies of TN and TP during EP- (CP-) El Niño. Seasonal study confirmed that water quality anomalies showed opposite patterns during EP- and CP-El Niño years. At the outlet of OTRB, seasonal anomalies in nutrients matched the El Niño-Southern Oscillation (ENSO) phases, illustrating the importance of climate variables associated with the two types of El Niño on water quality in the region. At the UMRB, TN and TP were also influenced by agriculture activities within the region and their anomalies became greater in the growing seasons during both EP- and CP-El Niño years. Quantitative analysis of precipitation, temperature, and their effects on nutrients suggested that precipitation played a more important role than temperature did in altering water quality in the Corn Belt region during both types of El Niño years. We also found specific watersheds (located in Iowa, Illinois, Minnesota, Wisconsin, and Indiana) that faced the greatest increases in TN and TP loads, were affected by both the precipitation and agricultural activities during the CP-El Niño years. The information generated from this study may help proper decision-making for water environment protection over the Corn Belt.



## 1 Introduction


The Corn Belt region of the United States (U.S.) primarily includes the Ohio-Tennessee River Basin (OTRB) and
the Upper Mississippi River Basin (UMRB) (Kellner and Niyogi, 2015; Panagopoulos et al., 2017; Ting et al., 2021).
The Corn Belt is a very important area of the agricultural activity of the country, which is also responsible for the
Gulf of Mexico hypoxic zone (Panagopoulos et al., 2014, 2015; Rabalais et al., 2007). The required nutrient
reduction of the region to decrease hypoxia is the highest among all regions in the Mississippi River Basin
(Mississippi River/Gulf of Mexico Watershed Nutrient Task Force, 2011). Hence, water quality change of the Corn
Belt region has been receiving considerable attention.
The El Niño-Southern Oscillation (ENSO) is a coupled ocean-atmosphere phenomenon that occurs across the
tropical Pacific (Trenberth, 1997; Wang and Kumar, 2015). Precipitation and temperature are influenced by ENSO
in many places in the U.S., including the Corn Belt region (Gershunov, 1998; Lee et al., 2014; Thomson et al., 2003;
Wang and Asefa, 2018). For example, more frequent dry conditions were found east of the Ohio River in the early
spring of a decaying El Niño (Wang and Asefa, 2018). ENSO events could also have a significant impact on the
water quality of a basin through changing climate factors. Heavy or prolonged rains might contribute to the pollutant
loading from agricultural runoff (Paul et al., 1997). Temperature anomalies might change river water quality by
affecting evaporation and water temperature. Keener et al. (2010) showed that ENSO significantly altered water
flow and nitrate concentration in a southeastern U.S. basin. Sharma et al. (2012) found that worse water quality in
southeast Alabama was linked to the stream temperature anomalies during El Niño years. However, what are the
impacts of ENSO on water quality in the Corn Belt region have not been studied.
In recent years, studies have found two different types of El Niño events: eastern Pacific (EP) and central Pacific
(CP) El Niños (Larkin and Harrison, 2005; Li et al., 2011; Tan et al., 2020; Tang et al., 2016; Yeh et al., 2009). The
former has warmer sea surface temperature anomalies (SSTAs) in the Niño 3 region (5 °N–5 °S, 150 °W–90 °W),
while the latter, also called El Niño Modoki, is manifested by maximum SSTAs in the Niño 4 region (5 °N–5 °S,
160 °E–150 °W). The effects of these two types of El Niño on regional climate and runoff are different. For example,
reduced rainfall was found in the northern, central, and eastern parts of the Amazon during the EP-El Niño years
(EP-ENYs), while increased rainfall anomalies were observed in most of the Amazon during the CP-El Niño years
(CP-ENYs) (Li et al., 2011). CP-El Niños were more effective in causing drought conditions in India due to
atmospheric subsidence than EP-El Niños (Kumar et al., 2006). How these two types of El Niño affect the water
quality of the Corn Belt has not been studied. Over recent years, EP-El Niño has appeared less frequently, whereas
CP-El Niño has become more common (Kao and Yu, 2009; Yeh et al., 2009). In the future, CP-El Niño will likely
happen more frequently (Yeh et al., 2009; Yu et al., 2010). Understanding the impact of these different El Niño
events on water quality over the Corn Belt is of critical importance for the water quality management of streams and
rivers.
In this study, we used the Soil and Water Assessment Tool (SWAT) model to estimate the water quality changes
due to EP- and CP-El Niños over the Corn Belt. The SWAT model is widely used to assess climate change and
alternative land use/land management scenarios on runoff and nutrients in a basin (Afonso de Oliveira Serrão et al.,
2022; Chaplot et al., 2004; Chen et al., 2021; Johnson et al., 2015; Vaché et al., 2002; Zhang et al., 2020). The





detailed objectives of this study were to 1) analyze the impacts of the two types of El Niño on water quality in the
Corn Belt region, and 2) identify the main climate factors that affect the change in water quality due to these El
Niños. Water quality change associated with future El Niño change was also discussed. Such information is
particularly important to enable decision-makers to take timely actions to reduce nutrient loading under climate
change.
**2 Data and methods**
**2.1 Data**
The study area includes two large basins in the U.S., namely, OTRB (528,000 km$^2$) and UMRB (492,000 km$^2$), as
shown in Fig. 1. OTRB comprises a significant portion of Pennsylvania, Ohio, West Virginia, Indiana, Illinois,
Kentucky, and Tennessee (Fig. 1a). The amount of annual rainfall in OTRB was high with an average of nearly
1200 mm during 1975–2016. OTRB's slopes are steep, especially in the forested Tennessee basin with slopes
greater than 5% in most (60%) of the area. The primary land use types are 50% forest, 20% cropland, and 15%
pasture (Fig. 1b). The cropland is mainly grown with corn, soybean, and wheat (Santhi et al., 2006). UMRB mainly
includes five states: Iowa, Illinois, Missouri, Wisconsin, and Minnesota (Fig. 1a). The mean annual value of rainfall
in UMRB during 1975–2016 was 900 mm. UMRB is relatively flat and most of the basin (75%) has a slope lower
than 5%. Cropland is the major land use type of the basin (50%) and is primarily grown with corn and soybean (Fig.
1b).
The weather data were obtained from 2,242 National Weather Service (NWS) stations in the study area. The
historical El Niño years were based on Table 1 in Li et al. (2011) and Table 2 in Ren et al. (2018). In summary, nine
EP-El Niño events (1976–1977, 1979–1980, 1982–1983, 1986–1987, 1987–1988, 1991–1992, 1997–1998, 2006–
2007, and 2015–2016) and six CP-El Niño events (1977–1978, 1990–1991, 1994–1995, 2002–2003, 2004–2005,
and 2009–2010) occurred during the study period (1975–2016). Significance was tested by the Monte Carlo method
(Levine and Wilks, 2000; Mo, 2010).
Available monthly streamflow and water quality data from 1975 to 2016 came from the 15 United States
Geological Survey (USGS) gaging stations in the study area. The final data used for calibration and validation of the
model were shown in Table 1.



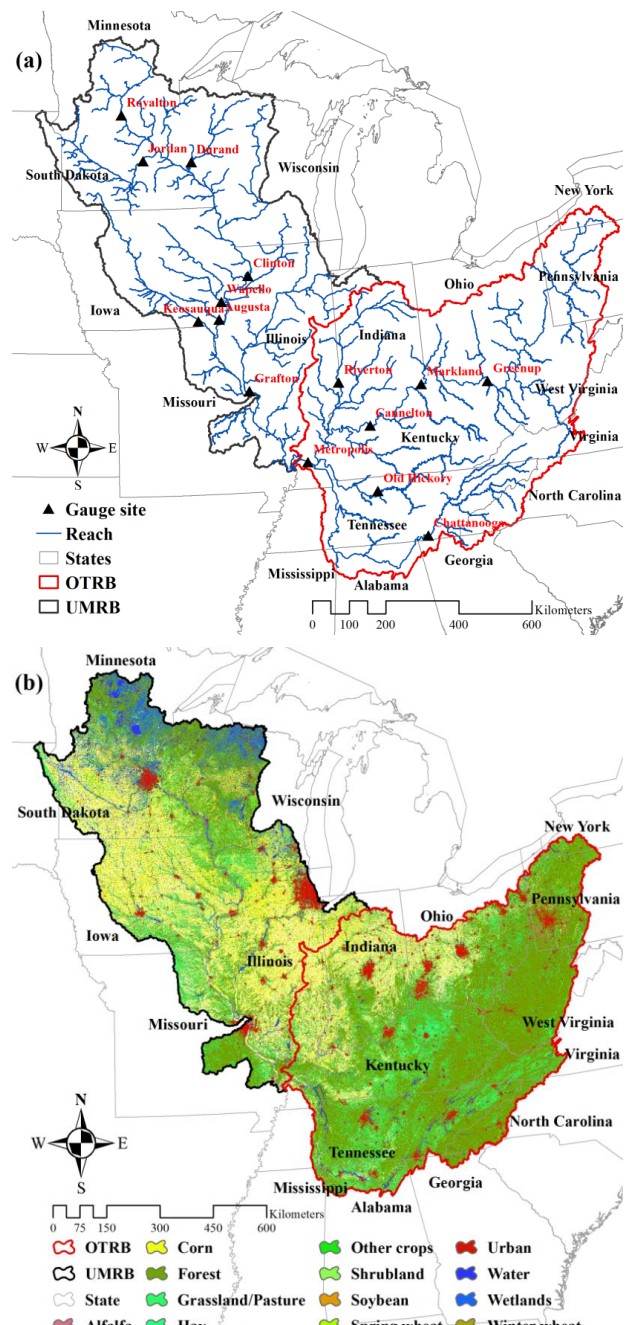


**Figure 1.** The Ohio-Tennessee River Basin (OTRB) and Upper Mississippi River Basin (UMRB) (a) available

United States Geological Survey (USGS) gage sites (black triangles), reaches (light blue), and the two watersheds

(heavy red: OTRB, heavy dark: UMRB) and (b) land use/land cover.





**Table 1.** Available periods of measured streamflow, total suspended sediment (TSS), total nitrogen (TN), and total
phosphorus (TP) at 15 USGS gauge at the OTRB and UMRB

| Site Name | Site Number | River Basin | Hydrologically Independent | Drainage (km$^2$) | Streamflow | TSS | TN | TP |
|---|---|---|---|---|---|---|---|---|
| Greenup | 03216600 | | Yes | 160,579 | 1975−2019 | - | 1975−2019 | 1975−2019 |
| Markland | 03277200 | | No | 215,409 | 1975−2019 | - | - | - |
| Riverton | 03342000 | | Yes | 34,087 | 1975−2019 | - | - | - |
| Old hickory | 03426310 | OTRB | Yes | 30,233 | 1988−2007 | - | - | - |
| Cannelton | 03303280 | | No | 251,229 | 1975−2019 | - | 1975−2019 | 1975−2019 |
| Metropolis | 03611500 | | No | 525,768 | 1975−2014 | - | 1975−2016 | 1975−2016 |
| Chattanooga | 03568000 | | Yes | 55,426 | 1975−2008 | - | - | - |
| Royalton | 05267000 | | Yes | 30,044 | 1975−2019 | - | - | - |
| Jordan | 05330000 | | Yes | 41,958 | 1975−2019 | - | - | - |
| Durand | 05369500 | | Yes | 23,336 | 1975−2019 | - | - | - |
| Clinton | 05420500 | | No | 221,703 | 1975−2019 | - | 1975−2019 | 1975−2019 |
| Augusta | 05474000 | UMRB | Yes | 11,168 | 1975−2019 | 1975−2017 | - | - |
| Wapello | 05465500 | | Yes | 32,375 | 1975−2019 | 1978−2017 | 1978−2019 | 1977−2019 |
| Keosauqua | 05490500 | | Yes | 36,358 | 1975−2019 | - | - | - |
| Grafton | 05587450 | | No | 443,665 | 1975−2019 | 1989−2017 | 1989−2019 | 1989−2019 |





**2.2 SWAT model description**
**2.2.1 Model description**
SWAT was developed by the U.S. Department of Agriculture Agricultural Research Services (Arnold et al., 1998). It has
been widely used in assessing the effects of climate and land use change on hydrological processes, sediment, and nutrients
in a basin (Neitsch et al., 2011; Pagliero et al., 2014; Yen et al., 2016). In the SWAT model, a basin is partitioned into sub-
basins, which are further divided into hydrological response units (HRUs) (Gassman et al., 2007; Neitsch et al., 2011;
Williams et al., 2008). Runoff, sediment, and nutrient loads are simulated for each HRU and then aggregated for the sub-
basins (Chen et al., 2021; Gassman et al., 2007; Neitsch et al., 2011). Hence, the pollution situation of each sub-basin during
different El Niño years could be obtained from this model.
**2.2.2 Model set-up and calibration**
In this study, 8-digit Hydrologic Unit Codes (HUC-8) defined by the USGS were selected as SWAT sub-basins. In total, the
OTRB and UMRB included 152 and 131 sub-basins, respectively. Flow paths between the sub-basins were determined using
the stream network of the National Hydrography Dataset Plus (NHDPlus) dataset developed by the USGS and U.S.
Environmental Protection Agency. Each of the sub-basins was further divided into several spatially uniform HRUs based on
land use, soil type, and slope (Chen et al., 2021; Neitsch et al., 2011). Thresholds of 0%, 10%, and 5% were used for land
use, soil, and slope, respectively, resulting in a total of 20,157 and 20,581 HRUs in the OTRB and UMRB. Then, point
sources (Schwarz et al., 2006), crop management (U.S. Department of Agriculture (USDA) - National Agricultural Statistics
Service (NASS), 2017), and tillage (Baker, 2011) dataset were incorporated to build the SWAT model (Chen et al., 2021).
SWAT Calibration and Uncertainty Programs (SWAT-CUP) with Sequential Uncertainty Fitting (SUFI-2) algorithm was
selected in this large-scale study to complete the calibration of the SWAT model (Abbaspour et al., 2012). The parameters of
water flow and water quality in OTRB and UMRB were selected based on a manual experimentation with SWAT parameters
and a literature review (Chen et al., 2021; Panagopoulos et al., 2014, 2015; Yen et al., 2016). The calibration steps followed
a recent study by Chen et al. (2021). The final parameters were shown in Table S1 and S2 in the supplementary materials.
The calibration results indicated that the SWAT model could rationally capture the observation (see Section 2.2.3).
**2.2.3 Model performance**
Overall, SWAT simulated the water flow of the OTRB and UMRB reasonably well in both calibration (1997–2016) and
validation (1975–1996) periods (Table 2). The coefficient of determination ($R^2$) and Nash-Sutcliffe efficiency (*NSE*) values
were larger than 0.5 for almost all the USGS gages except Chattanooga, and percent bias (*PBIAS*) values were all acceptable
during the calibration periods (1997–2016) based on the ≤ ±25% deviation criterion (Moriasi et al., 2015). The validation
results also showed acceptable static values and in some cases were even better, such as Chattanooga and Grafton. Figures





S1 and S2 in the supplementary materials further demonstrated good agreement between the calculated and observed
streamflow across OTRB and UMRB; particularly, most of the peaks and recession limbs were well matched in the
simulations.
Modeled water quality was generally in agreement with the observations for the two river basins, as most of the *PBIAS*
values were within bias criteria for sediment, TN, and TP during both the calibration and validation periods (Santhi et al.,
2014) (Table 3). The only exception that did not meet the criteria was the sediment simulation at Augusta. The upstream
drainage area of Augusta was relatively small (only around 3% of the UMRB), thus its influence on downstream sediment
and pollutant transport downstream was minor (Fig. 1a). The sediment statistics in OTRB were not calculated here because
of a lack of observations (Panagopoulos et al., 2015). Instead, we compared the simulated annual sediment at Metropolis
with observations during 1975–2010 and found that the difference was also within the bias criteria for sediment
(Panagopoulos et al., 2015; Santhi et al., 2014). Moreover, most of the *NSE* and $R^2$ values were positive and many $R^2$ values
were greater than 0.5, indicating that the simulated water quality was reasonable (Chen et al., 2021). This finding could be
further proved by the graphs of observed and simulated sediment and nutrients (Figs. S3 and S4).
**Table 2.** Monthly streamflow calibration and validation statistics.

| Gauge Site | Calibration (1997−2016) | | | Validation (1975−1996) | | |
|---|---|---|---|---|---|---|
| | $R^2$ | *NSE* | *PBIAS* | $R^2$ | *NSE* | *PBIAS* |
| Greenup | 0.88 | 0.87 | 1.6 | 0.87 | 0.82 | 15.9 |
| Markland | 0.88 | 0.87 | 6.4 | 0.88 | 0.81 | 17.4 |
| Riverton | 0.84 | 0.83 | −1.4 | 0.8 | 0.78 | −7.4 |
| Old Hickory | 0.82 | 0.75 | −4.7 | 0.81 | 0.79 | 7.8 |
| Cannelton | 0.89 | 0.87 | 9.6 | 0.9 | 0.84 | 15.9 |
| Metropolis | 0.84 | 0.83 | 4.1 | 0.88 | 0.8 | 15.8 |
| Chattanooga | 0.59 | 0.48 | 6.1 | 0.62 | 0.53 | 10.7 |
| Royalton | 0.63 | 0.56 | −4.7 | 0.58 | 0.55 | −0.5 |
| Jordan | 0.64 | 0.59 | 0.7 | 0.77 | 0.66 | −22.8 |
| Durand | 0.65 | 0.5 | −20.9 | 0.6 | 0.53 | −10.7 |
| Clinton | 0.63 | 0.56 | 6.6 | 0.63 | 0.59 | 9.8 |
| Augusta | 0.75 | 0.7 | −9.9 | 0.78 | 0.76 | −9.9 |
| Wapello | 0.71 | 0.67 | −2.3 | 0.76 | 0.76 | −1 |
| Keosauqua | 0.76 | 0.73 | 5.8 | 0.8 | 0.78 | 14 |
| Grafton | 0.69 | 0.62 | 12.5 | 0.77 | 0.7 | 16.1 |









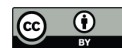


**Table 3.** Monthly TSS, TN, and TP calibration and validation statistics.

| Variable | Gauge Site | Calibration(1997−2016) | | | Validation(1975−1996) | | |
|---|---|---|---|---|---|---|---|
| | | $R^2$ | NSE | PBIAS | $R^2$ | NSE | PBIAS |
| TSS | Augusta | 0.34 | 0.1 | 64.5 | 0.39 | 0.02 | 77.2 |
| | Wapello | 0.45 | 0.15 | −21.5 | 0.51 | 0.48 | 13.6 |
| | Grafton | 0.43 | 0.22 | 0.7 | 0.26 | 0.07 | 13.9 |
| TN | Greenup | 0.57 | 0.23 | −13.7 | 0.59 | 0.52 | 17 |
| | Cannelton | 0.63 | 0.59 | 8.3 | 0.59 | 0.46 | 26.9 |
| | Metropolis | 0.58 | 0.36 | −9.2 | 0.57 | 0.51 | 14.9 |
| | Clinton | 0.43 | −0.61 | 2.7 | 0.36 | −0.25 | 9.4 |
| | Wapello | 0.51 | 0.05 | 1.8 | 0.44 | 0.32 | 17.6 |
| | Grafton | 0.54 | 0.15 | 2.2 | 0.53 | 0.08 | 2 |
| TP | Greenup | 0.56 | 0.46 | −29.6 | 0.59 | 0.57 | 8.6 |
| | Cannelton | 0.56 | 0.48 | 21.2 | 0.44 | 0.35 | 28.1 |
| | Metropolis | 0.49 | 0.41 | −8.6 | 0.45 | 0.38 | −5.2 |
| | Clinton | 0.44 | −0.59 | −11.1 | 0.42 | 0.13 | 10.9 |
| | Wapello | 0.6 | 0.24 | −19.1 | 0.55 | 0.29 | −16.3 |
| | Grafton | 0.57 | 0.25 | 0.8 | 0.54 | 0.12 | −14.8 |

**3 Results**
**3.1 Impacts of the EP- and CP-El Niños on the water quality in the Corn Belt**
**3.1.1 Annual composite**
(1) Water quality at the outlet
Table 4 lists the nutrient change during EP- and CP-ENYs at the outlet of OTRB and UMRB. Annual loads of TN and TP
decreased during the EP-ENYs, while the pattern reversed during the CP-ENYs in the U.S. Corn Belt region. Specifically,
compared to normal years, the TN and TP decreased by 13.1% (61,300 metric ton yr$^{-1}$, hereafter ton yr$^{-1}$) and 14.0% (7,300
ton yr$^{-1}$) during EP-ENYs, respectively, whereas they increased by 3.3% (15,500 ton yr$^{-1}$) and 4.6% (2,400 ton yr$^{-1}$) during
CP-ENYs at the outlet of the OTRB, respectively (Table 4). TN and TP at the outlet of the UMRB showed a similar pattern
as that of the OTRB, decreasing (increasing) by 18.5% (5.7%) and 19.8% (4.4%) in EP (CP)-ENYs. Furthermore, EP-El
Niños had a much greater impact on water quality than CP-El Niños at the outlets of OTRB and UMRB. The magnitudes of
variation in both TN and TP during the EP-ENYs were three to four times greater than those during the CP-ENYs (Table 4).







**Table 4.** Soil and Water Assessment Tool (SWAT) model's estimates of annual mean TN and TP at the outlets of the OTRB and UMRB during all simulation year (1975−2016), eastern Pacific (EP)-El Niño years and central Pacific (CP)-El Niño years.

| River Basin | Variable | Average | EP | | CP | |
|---|---|---|---|---|---|---|
| | | | Anomaly | Percent (%) | Anomaly | Percent (%) |
| OTRB | TN ($\times 10^3$ ton yr$^{-1}$) | 469.0 | −61.3 | −13.1 | 15.5 | 3.3 |
| | TP ($\times 10^3$ ton yr$^{-1}$) | 52.2 | −7.3 | −14.0 | 2.4 | 4.6 |
| UMRB | TN ($\times 10^3$ ton yr$^{-1}$) | 526.2 | −97.5 | −18.5 | 29.9 | 5.7 |
| | TP ($\times 10^3$ ton yr$^{-1}$) | 41.0 | −8.1 | −19.8 | 1.8 | 4.4 |

(2) Water quality at the sub-basin scale

We analyzed water quality change on the sub-basin scale during EP-ENYs and CP-ENYs, respectively (Fig. 2). Anomalous patterns of water quality associated with El Niño events within the Corn Belt varied in space. Clearly, more sub-basins showed negative anomalies of TN and TP during EP-ENYs, whereas more sub-basins showed positive anomalies during CP-ENYs (Fig. 2). Specifically, during EP-ENYs, significant below-average TN and TP were found almost in the whole OTRB and UMRB with maximum reductions of TN and TP up to −11.7 and −0.9 kg ha$^{-1}$, respectively, which were of similar magnitudes to the mean values (12.7 kg of N ha$^{-1}$ and 1.0 kg of P ha$^{-1}$) in the Corn Belt region (Figs. 2a and 2b). During CP-ENYs, positive anomalies mainly occurred throughout the southern OTRB and UMRB. In the northern UMRB and OTRB, about 42.4% and 41.7% of sub-basins tended to have below-average TN and TP (Figs. 2c and 2d). These patterns coincided with the TN and TP changes at the outlets of the two basins, which could also explain why greater changes in TN and TP occurred in EP-ENYs than in CP-ENYs at the outlets of the OTRB and UMRB (Table 4).

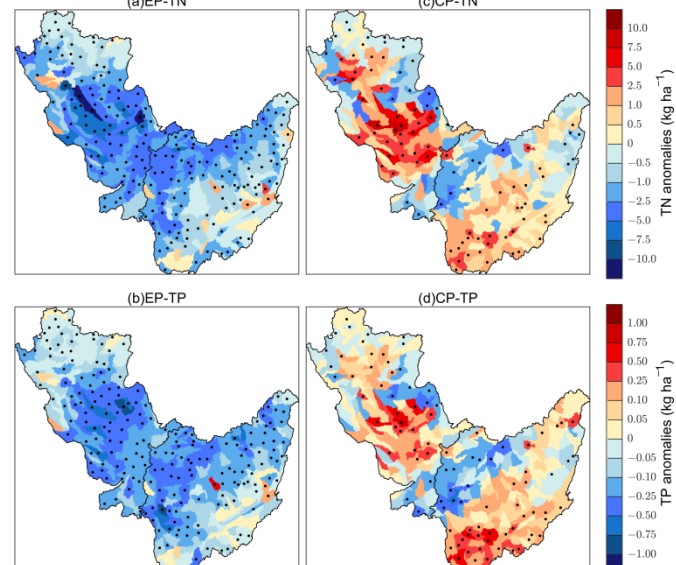

186

**Figure 2.** Composite results of annual TN and TP anomalies (unit: kg ha$^{-1}$) in EP-El Niño years (a and b) and in CP-El Niño

years (c and d) during the period of 1975–2016. Stippling denotes anomalies significantly different from zero at the 95%

confidence level based on the Monte Carlo test.

190

### 3.1.2 Seasonal composite

(1) Water quality at the outlet

Figure 3a showed TN anomalies at the OTRB and UMRB outlets in each season during EP- and CP-ENYs. At the outlet of

the OTRB, Seasonal anomalies of the water quality reached the maximum when ENSO signals were the strongest (Fig. 3a).

El Niño usually developed in boreal summer (June-August, JJA) and autumn (September-November, SON), peaked in

winter (December of the current year and January and February of the following year, DJF), and decayed in spring (March-

May, MAM) (Trenberth, 1997; Li et al., 2011). Maximum changes in TN occurred in the winter and spring seasons during

EP-ENYs (decreased by $28.3 \times 10^3$ and $29.5 \times 10^3$ metric tons (hereafter ton), respectively, Fig. 3a). During CP-ENYs, TN

increased by $11.7 \times 10^3$ tons in winter, and did not change much in the rest of the three seasons (spring, summer, and autumn)

compared to that in the normal years (Fig. 3a). Seasonal TN anomalies at the outlet of the UMRB were different from those

of the OTRB. Figure 3a showed that TN decreased by $71.0 \times 10^3$ and $30.8 \times 10^3$ tons, respectively, in the spring and summer

seasons during EP-ENYs; but in winter and autumn, TN did not change much compared to normal years. Similarly, during

CP-ENYs, TN anomalies were greater in spring and summer although TN anomalies became positive during CP-ENYs,

different from TN changes during EP-ENYs (Fig. 3a).



Figure 3b demonstrated that the seasonal changes of TP during EP- and CP-ENYs were similar to those of TN in both
OTRB and UMRB during EP-ENYs. At the OTRB the magnitudes of TP reduction in boreal winter and spring were greater
than those in the summer and autumn seasons during EP-ENYs. During CP-ENYs, TP anomalies were greater in both
autumn and winter (Fig. 3b). This phenomenon was probably related to the different duration of the two types of El Niño.
The mean duration of EP-El Niño was about 15 months (Mo, 2010), its impact on water quality could last into the following
spring; while El Niño Modoki usually lasted for about eight months (Mo, 2010, Yu et al., 2010), therefore the impacts of the
CP-El Niño on water quality usually ended in the winter. Besides, at the UMRB maximum changes in TP occurred in the
spring and summer seasons during both EP- and CP-ENYs.
In summary, more seasons showed negative anomalies of water quality during EP-ENYs, whereas more seasons showed
positive anomalies during CP-ENYs (Fig. 3). Consequently, the annual TN and TP anomalies over the Corn Belt region
showed the opposite pattern during EP-ENYs and CP-ENYs (Table 4).

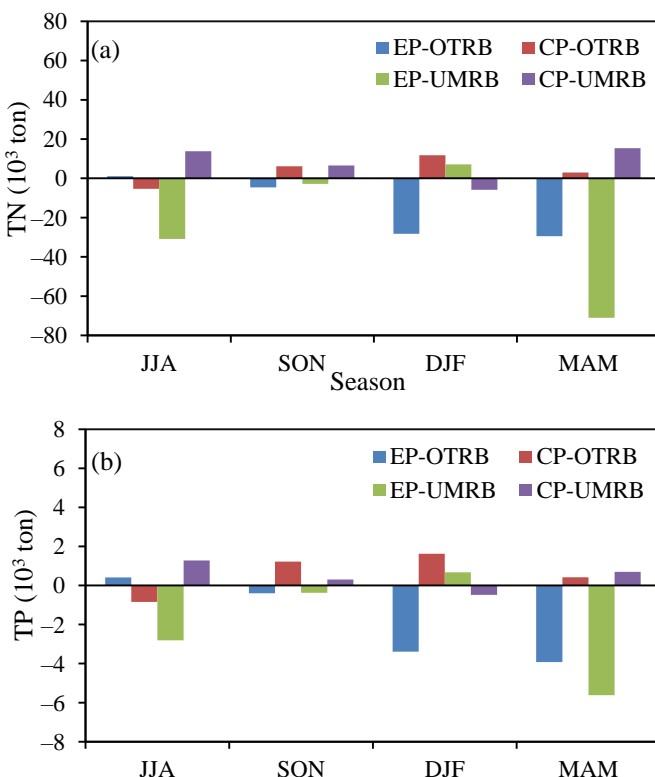


**Figure 3.** Seasonal anomalies of (a) TN and (b) TP (unit: $10^3$ tons) in summer (June-August, JJA), autumn (September-
November, SON), winter (December of the current year and January and February of the following year, DJF) and spring
(March-May, MAM) at the outlets of the OTRB and UMRB during EP-El Niño years and CP-El Niño years.






(2) Water quality at the sub-basin scale
Water quality at the Corn Belt can vary in different locations/sub-basins and change between seasons. Figure 4 showed
spatial patterns of TN and TP anomalies at the OTRB and UMRB in each season during the CP- and EP-ENYs, separately.
EP-El Niño was characterized by negative TN and TP anomalies over most of the OTRB and UMRB for all seasons (Figs.
4a–4d and 4i–4l). Significant below-average TN and TP occurred in almost all of the UMRB and the eastern OTRB in the
summer when EP-El Niño was developing in the tropical Pacific, with maximum reductions up to −4.5 and −0.45 kg ha$^{-1}$,
respectively (Figs. 4a and 4i). The negative water quality anomalies moved to the southern UMRB and northern OTRB in
the autumn (Figs. 4b and 4j). These negative anomalies further moved to the whole OTRB when EP-El Niño was mature in
the winter (Figs. 4c and 4k). This result generally agreed with previous findings of the El Niño impacts on the precipitation
in the study area, which indicated that the precipitation in the Ohio Valley was sensitive to El Niño events and showed
negative precipitation anomalies at EP-ENYs (Mo, 2010; Twine et al., 2005). In spring, severe TN and TP deficits were most
apparent almost all over the Corn Belt area (Figs. 4d and 4l).

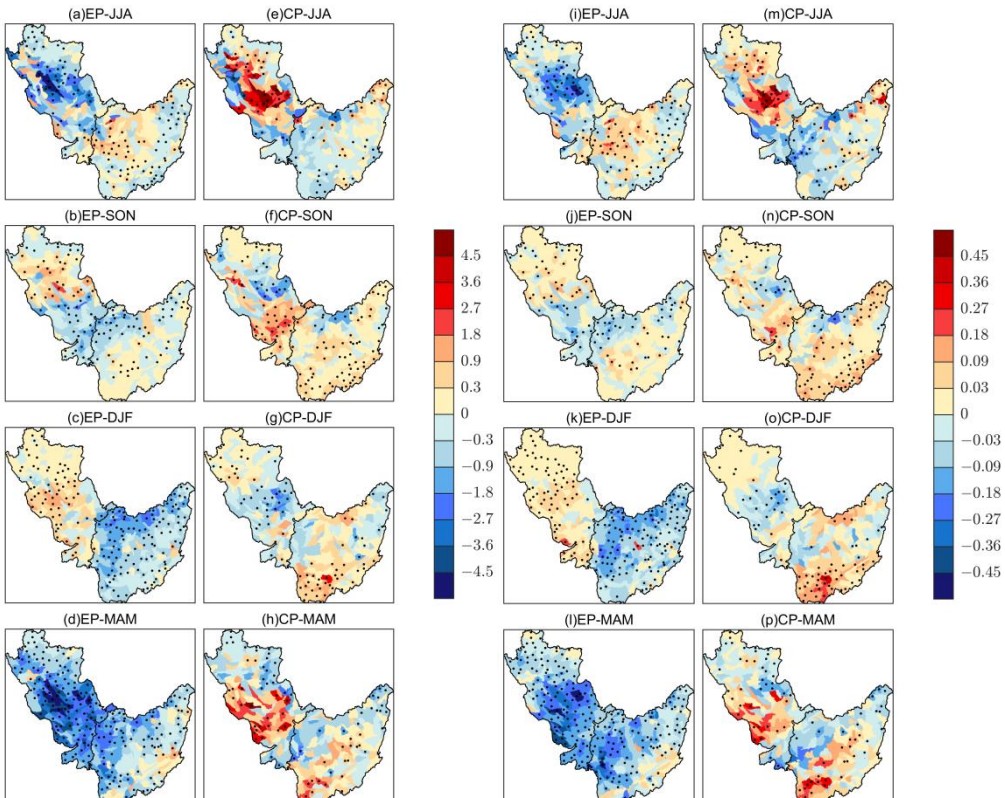


**Figure 4.** Seasonal composite of (a-h) TN and (i-p) TP anomalies (unit: kg ha$^{-1}$) in summer (JJA) (a and i), autumn (SON)
(b and j), winter (DJF) (c and k), and spring (MAM) (d and l) in EP-El Niño years during the period of 1975–2016; (e-h) and
(m-p) are the same as (a-d) and (i-l) but for CP-El Niño years. Stippling denotes anomalies significantly different from zero
at the 95% confidence level based on the Monte Carlo test.



In contrast, during CP-ENYs, positive TN and TP anomalies were scattered in most of the Corn Belt region with the
highest TN and TP anomalies increase up to 4.5 and 0.45 kg ha$^{-1}$, respectively (Figs. 4e–4h and 4m–4p). In summer,
northern UMRB and eastern OTRB were characterized by above-normal water quality (Figs. 4e and 4m). The positive TN
and TP anomalies moved to the southern UMRB and whole OTRB in the autumn (Figs. 4f and 4n). In winter, these positive
anomalies were concentrated in the southern OTRB and northern UMRB (Figs. 4g and 4o). In the spring, abnormally high
TN and TP mainly occurred in the southern part of the UMRB while most of the positive and negative anomalies in the
OTRB region were insignificant (Figs. 4h and 4p).
In conclusion, water quality anomalies showed opposite patterns during EP-ENYs and CP-ENYs on both annual and
seasonal time scales in the Corn Belt region. Furthermore, EP-El Niño seemed to have a greater and long-lasting impact on
TN and TP than CP-El Niño. Hence, treating the two as a single phenomenon was not appropriate when analyzing the
impacts of ENSO on water quality.
**3.2 Possible climate reasons for the water quality change during CP- and EP-El Niño events**
As precipitation and temperature usually respond differently to the two types of El Niño at different temporal and spatial
scales (Li et al., 2011; Tan et al., 2020), we hereafter analyzed these climate factors' impacts accordingly in the Corn Belt
region.
**3.2.1 Precipitation**
Decreased precipitation in EP-ENYs was one of the important reasons that improved water quality in the Corn Belt region.
This finding could be shown in the spatial patterns of annual precipitation (Fig. 5a), and TN and TP anomalies (Figs. 2a and
2b). For example, significantly below-normal precipitation occurred in much of the OTRB and UMRB (278 out of 283 sub-
basins) during EP-ENYs (Fig. 5a), thus TN and TP were reduced in most parts of the Corn Belt (Figs. 2a and 2b). During
CP-ENYs, precipitation anomalies became positive in 226 out of 283 sub-basins (Fig. 5c). Correspondingly, TN and TP
concentrations were elevated compared to normal years (Figs. 2c and 2d). Spatially, there were more sub-basins (98.2%) and
larger area with negative precipitation anomalies in EP-ENYs than positive anomalies (79.9% sub-basins) during CP-ENYs
in the Corn Belt, indicating that EP-El Niño tended to have a much wider impact on precipitation and thus water quality than
CP-El Niño.
To better understand how precipitation affected water quality in the Corn Belt, variations of annual runoff during EP-
ENYs and CP-ENYs were also discussed because nitrogen and phosphorus were transported by runoff (Neitsch et al., 2011),
and precipitation was a very important source of runoff (Gassman et al., 2007). The overall patterns of runoff anomalies
(Figs. 5b and 5d) were similar to those of precipitation anomalies (Figs. 5a and 5c) during the two types of El Niño with
pattern correlations being 0.79 at OTRB and 0.75 at UMRB in EP-ENYs, 0.96 and 0.77 in CP-ENYs, respectively.
Specifically, during EP-ENYs, negative annual runoff anomalies occurred in most of the Corn Belt region (Fig. 5b),
resulting in reduced TN and TP compared to normal years (Figs. 2a and 2b). In contrast, in CP-ENYs, positive runoff

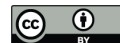

anomalies were mainly concentrated in southern OTRB and UMRB (Fig. 5d), more nutrients were thus carried out by the
runoff associated with above-normal precipitation in the area (Figs. 2c and 2d). We also noticed that EP-El Niño had a wider
influence on runoff than CP-El Niño, as 71 more sub-basins (259 vs 188) and larger areas with significant runoff anomalies
were found in EP-ENYs than in CP-ENYs (Figs. 5b and 5d), generally consistent with the spatial patterns of precipitation
changes due to the El Niños (Figs. 5a and 5c). The phenomena could partly explain why EP-El Niños tended to have greater
impacts on the water quality than CP-El Niños at the outlets of the OTRB and UMRB.

Some differences also existed among precipitation, runoff, and nutrients. For precipitation and runoff, annual precipitation

anomalies were greater than runoff anomalies in the El Niño years. For example, precipitation anomalies of many sub-basins
in northern UMRB were stronger than $-0.2$ mm day$^{-1}$ during EP-ENYs, but the magnitudes of runoff anomalies were
weaker than $-0.2$ mm day$^{-1}$ (Figs. 5a and 5b). During CP-ENYs, most annual precipitation and runoff anomalies were
positive, but the magnitudes of precipitation anomalies were generally higher than that of runoff anomalies (Figs. 5c and 5d).
These findings suggested that the impact of El Niño on runoff was weakened by the land surface hydrological process. For
runoff and nutrients, changes of runoff in El Niño years were greater in the OTRB than in the UMRB (Figs. 5b and 5d),
whereas changes of TN and TP were smaller in the OTRB than in the UMRB during both EP-ENYs and CP-ENYs (Fig. 2).
The discrepancies between annual runoff and nutrient anomalies in the OTRB and UMRB may be related to the presence of
more cropland in the UMRB where more fertilizers were used (Chen et al., 2021, see Section 4.1 for details), hence a
relatively small change of runoff due to El Niño-induced precipitation could lead to large TN and TP variations.

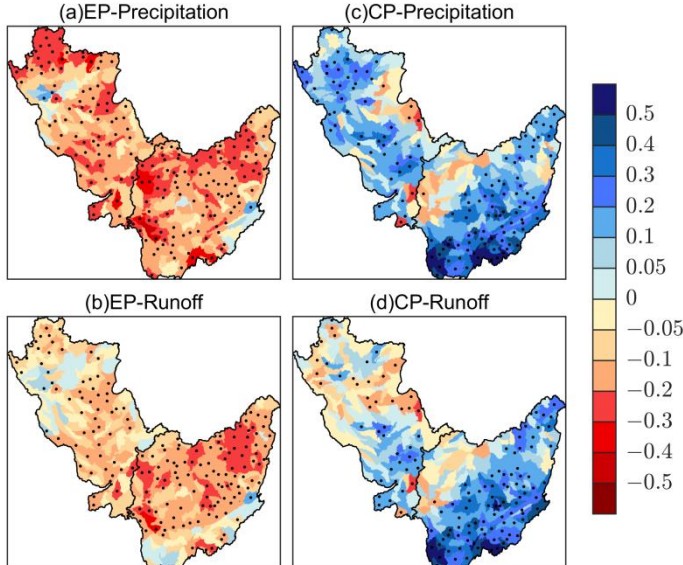


**Figure 5.** Composite of annual precipitation and runoff anomalies (unit: mm day$^{-1}$) for the two types of El Niño during the
period of 1975–2016 (a and b) in EP-El Niño years and (c and d) in CP-El Niño years. Stippling denotes anomalies
significantly different from zero at the 95% confidence level based on the Monte Carlo test.



Seasonal patterns of precipitation and runoff anomalies (Fig. 6) further proved the impacts of precipitation on water
quality through runoff during EP-ENYs and CP-ENYs. In summer, significantly below-normal precipitation occurred in
almost all of the UMRB when EP-El Niño was developing in the tropical Pacific, with maximum reductions of precipitation
up to $-0.9$ mm day$^{-1}$ in the region (Fig. 6a). The runoff anomaly pattern was much the same as that of precipitation, but in a
weaker magnitude—less than $-0.6$ mm day$^{-1}$ in UMRB (Fig. 6i). At the same time, TN and TP decreased in the area (Figs.
4a and 4i) because fewer nutrients were carried out by the reduced runoff in summer (Fig. 6i). Similar change patterns of
precipitation (Figs. 6b-6d), runoff (Figs. 6j-6l), and nutrients (Figs. 4b-4d and 4j-4l) could also be found in other seasons
during EP-ENYs. We noticed that negative runoff and precipitation anomalies reached their maximum in spring throughout
the Corn Belt, leading to better water quality in the region compared to the normal years. CP-El Niño events caused the
opposite patterns of seasonal precipitation (Figs. 6e-6h) and runoff anomalies (Figs. 6m-6p) in the Corn Belt region; TN and
TP thus increased in central and southern UMRB in the spring, summer, and autumn, and most of OTRB from autumn to
winter (Figs. 4e-4h and 4m-4p). Some differences also existed between precipitation and water quality in the UMRB,
especially in spring and summer. During these two seasons, the variation of the seasonal precipitation at each sub-basin was
relatively uniform in both EP-ENYs (Figs. 6a and 6d) and CP-ENYs (Figs. 6e and 6h), but nutrient variations were high in
some sub-basins of the UMRB (Figs. 4a, 4d, 4e, 4h, 4i, 4l, 4m, and 4p). This phenomenon suggested that water quality was
also influenced by local factors besides climate variables associated with El Niños.

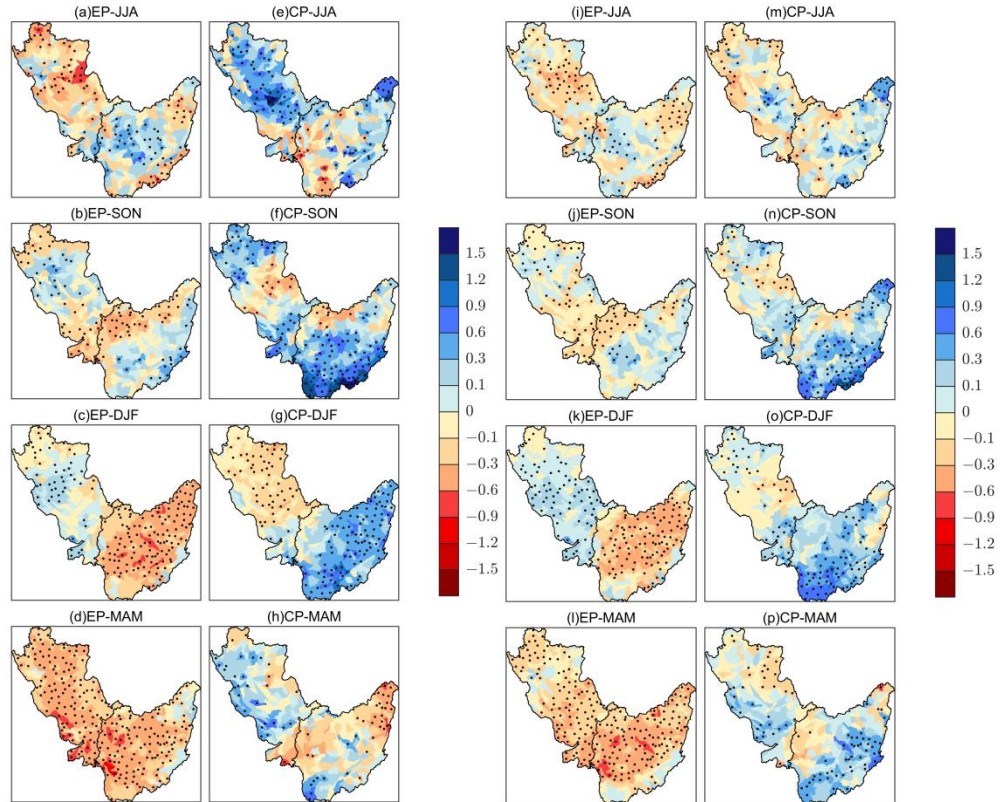

**Figure 6.** Composite of seasonal (a-h) precipitation and (i-p) runoff anomalies (unit: mm day$^{-1}$) in JJA (a and i), SON (b and j), DJF (c and k), and MAM (d and l) in EP-El Niño years during the period of 1975−2016; (e-h) and (m-p) are the same as (a-d) and (i-l) but for CP-El Niño years. Stippling denotes anomalies significantly different from zero at the 95% confidence level based on the Monte Carlo test.

### 3.2.2 Temperature

Temperature changes have previously been found to affect water quality by changing the evaporation process of the water cycle (Neitsch et al., 2011 Sun et al., 2011). Thus, how evaporation varied associated with temperature change during El Niños in the Corn-Belt region was analyzed. Compared to normal years, the annual temperature increased over the UMRB, but decreased in the OTRB, especially in the southern OTRB during EP-ENYs (Fig. 7a). Evaporation slightly increased in most of OTRB (Fig. 7b), which did not share the same pattern with temperature change on the annual time scale (Fig. 7a). This might be due to the fact that temperature directly affected potential evapotranspiration (Neitsch et al., 2011), the ability of the atmosphere to remove water from the surface through both evaporation and transpiration, but the actual evaporation/evapotranspiration was also related to other variables such as the amount of water available for evaporation besides temperature. Enhanced evaporation further reduced runoff (Bales et al., 2017). Thus, decreased precipitation (Fig. 5a)



and enhanced evaporation (Fig. 7b) during the EP-ENYs would facilitate runoff decline and cause a much wider impact of
EP-El Niño events on water quality in the Corn Belt region (Fig. 2). During CP-ENYs, temperature decreased insignificantly
in most of the Corn Belt region (Fig. 7c). Evaporation increased in more sub-basins over the UMRB (Fig. 7d). The enhanced
evaporation (Fig. 7d) tended to offset, to some extent, the impact of higher than normal precipitation (Fig. 5c) on water
quality during CP-ENYs.
The impacts of the two climate factors, precipitation and temperature (through evaporation), on runoff were compared.
During the El Niño years, the magnitude of annual precipitation change was often greater than 0.1 mm day$^{-1}$ (Figs. 5a and 5c)
while most of the annual evaporation varied between -0.05 and 0.05 mm day$^{-1}$ (Figs. 7b and 7d). This suggested that both
precipitation and evaporation influence water quality through runoff, but precipitation seemed to play a more important role
in altering water quality over the Corn Belt region during El Niño years.




**Figure 7.** Composite results of annual average temperature (a and c, unit: °C) and evaporation (b and d, unit: mm day$^{-1}$)
anomalies for EP-El Niño years (a and b) and CP-El Niño years (c and d), respectively. Stippling denotes anomalies
significantly different from zero at the 95% confidence level based on the Monte Carlo test.

Figure 8 showed seasonal patterns of temperature and evaporation anomalies during the CP- and EP-ENYs. In EP-ENYs,
significantly below-normal temperature occurred throughout the OTRB in the summer (Fig. 8a) and expanded to the entire
Corn Belt region in the autumn (Fig. 8b). In winter and spring, significantly positive temperature anomalies were shown in
the UMRB and most of OTRB (Figs. 8c and 8d). Corresponding to the seasonal temperature anomalies, evaporation varied
differently at different seasons (Figs. 8i-8l). In the summer and autumn, most sub-basins had negative evaporation anomalies
with decreased temperature (Figs. 8i and 8j); but in winter and spring, significantly above-normal evaporation occurred with
increased temperature in EP-ENYs (Figs. 8k and 8l). During CP-ENYs, the summer season was characterized by negative
temperature anomalies throughout the Corn Belt region, with the maximum anomalies up to −1.2 ℃ (Fig. 8e). Evaporation
anomalies became negative in most sub-basins (Fig. 8m). The temperature pattern was reversed in the autumn, with positive
temperature and evaporation anomalies over the entire region (Figs. 8f and 8n). In winter, temperature anomalies became
negative again (Fig. 8g), evaporation also reduced, with significantly negative anomalies in the OTRB and central and
southern UMRB (Fig. 8o). Temperature anomalies in spring were insignificant (Fig. 8h) most likely due to the short eight-
month duration of the CP-El Niño (Mo, 2010, Yu et al., 2010), although positive evaporation anomalies appeared in the
northern UMRB (Fig. 8p).

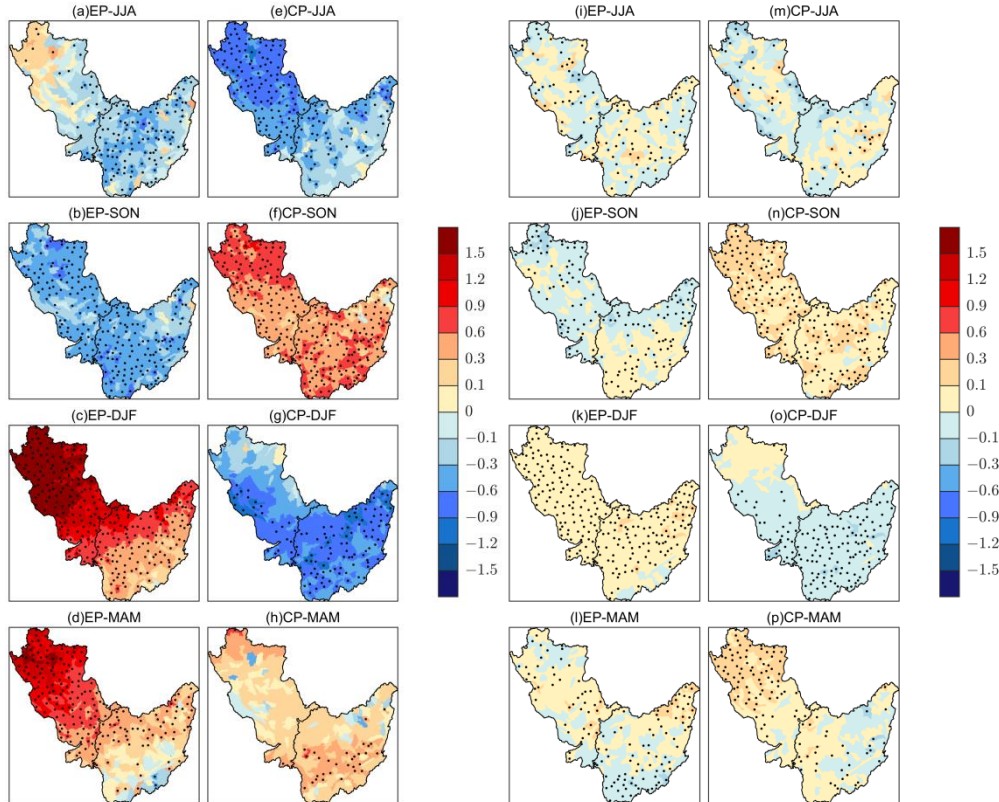


**Figure 8.** Composite results of seasonal temperature (a-h, unit: °C) and evaporation (i-p, unit: mm day$^{-1}$) anomalies in JJA (a
and i), SON (b and j), DJF (c and k), and MAM (d and l) in EP-El Niño years (a-d, i-l) and CP-El Niño years (e-h, m-p),
respectively. Stippling denotes anomalies significantly different from zero at the 95% confidence level based on the Monte
Carlo test.



**4 Discussion**
**4.1 Agricultural activity**
This study focused on climate impact on water quality during EP-ENY and CP-ENY, we did not perform sensitivity
experiments on agriculture activities. In the SWAT model runs, the distribution of agriculture activities pattern (2008
Cropland Data Layers (CDL); USDA-NASS, 2016) was kept the same during El Niño and normal years. Corresponding to
the described agriculture activities, the corn growing areas, i.e., the southern UMRB and northern OTRB, usually produced
greater annual TN loads (>10 kg of N ha$^{-1}$) and TP loads (>1 kg of P ha$^{-1}$) (Fig. S5). During EP-ENYs, the nutrients were
largely reduced (>1 kg of N ha$^{-1}$ and >0.1 kg of P ha$^{-1}$) in the two corn-growing regions because of decreased precipitation
($< -0.1$ mm day$^{-1}$) (Figs. 2a, 2b, and 5a). In CP-ENYs, the nutrient level increased in the southern UMRB (Figs. 2c and 2d)
since enhanced precipitation in CP-ENYs exacerbated the water quality in the area of heavy agriculture activities (Fig. 5c).
Water quality in the OTRB region showed different change patterns from the agriculture activities, i.e., TN and TP decreased
in the northern OTRB but increased in the southern OTRB (Figs. 2c and 2d). Such changes in water quality followed the
precipitation change in the OTRB (Fig. 5c), demonstrating that CP-El Niño-induced precipitation change played a more
important role in modulating water quality in OTRB.
On seasonal scales, changes in nutrients' magnitudes were stronger in spring and summer, especially in UMRB (Fig. 4).
This pattern was also related to the growth period of crops in the Corn Belt. The major crops here are corn and soybean,
which are often planted and fertilized in May and harvested in October. Hence, nutrients might be more likely to be removed
from the soil during spring and summer.
**4.2 Equivalent impacts of CP- and EP-El Niño on water quality in specific watersheds**
Section 3 discussed the impacts of El Niño on water quality at the outlets and sub-basin scales. At the outlets, EP-El Niño
had a much greater impact on TN and TP than CP-El Niño both at annual and seasonal time scales. But at the sub-basin scale,
CP- and EP-El Niño could have equivalent impacts on water quality in specific watersheds, predominantly in Iowa (IA),
Illinois (IL), Minnesota (MN), Wisconsin (WI), and Indiana (IN), which contributed the greatest amounts of nutrient change
to the whole basin loads (Fig. 2). Table 5 listed the top 10 HUC-8 sub-basins with the largest nutrient change during the two
types of El Niño years. In EP-ENYs, TN anomalies changed from 6.2 and 11.7 kg ha$^{-1}$ among the top 10 HUC-8 sub-basins;
while in CP-ENYs, TN anomalies changed from 5.6 and 9.3 kg ha$^{-1}$ (Table 5). These changes in TN during classic El Niño
and El Niño Modoki were comparable. Analysis of the top 10 HUC-8 sub-basins with the largest TP change during the EP-
and CP-El Niño illustrated similar results (not shown). These findings indicated that CP-El Niño could have comparable
impacts on TN and TP as EP-El Niño at the hot spot sub-basins although EP-El Niño had a much broader and longer impact
on water quality at the outlets.





**Table 5.** Information on the top 10 8-digit Hydrologic Unit Codes (HUC-8) sub-basins with greatest TN anomalies during
EP- and CP-El Niño years.

| El Niño Type | HUC-8 | TN (kg ha$^{-1}$) | | Precipitation (mm yr$^{-1}$) | | Cropland Percentage (%) | States |
|---|---|---|---|---|---|---|---|
| | | Average | Anomaly | Average | Anomaly | | |
| EP | 07080102 | 51.2 | −11.7 | 912.9 | −54.8 | 70.1 | IA,MN |
| | 07080201 | 33.7 | −10.8 | 909.9 | −51.1 | 73.5 | IA,MN |
| | 07090006 | 37.8 | −10.4 | 935.2 | −65.7 | 68.8 | IL,WI |
| | 07020011 | 28.3 | −8.5 | 845.1 | −84.0 | 78.6 | MN |
| | 07080202 | 27.4 | −7.5 | 890.2 | −32.9 | 72.1 | IA,MN |
| | 05120107 | 24.8 | −7.2 | 1068.8 | −102.2 | 73.8 | IN |
| | 07090007 | 31.2 | −6.9 | 949.4 | −87.6 | 78.9 | IL |
| | 07080204 | 33.0 | −6.6 | 918.0 | −69.4 | 76.0 | IA |
| | 07080208 | 27.0 | −6.3 | 904.4 | −62.1 | 59.9 | IA |
| | 07080209 | 25.6 | −6.2 | 914.8 | −94.9 | 58.4 | IA |
| CP | 07080103 | 26.2 | 9.3 | 948.0 | 142 | 70.2 | IA |
| | 07080208 | 27.0 | 9.1 | 904.4 | 135 | 59.9 | IA |
| | 07080206 | 21.9 | 9.1 | 923.1 | 128 | 63.3 | IA |
| | 07090007 | 31.2 | 7.3 | 949.4 | 62 | 78.9 | IL |
| | 07080203 | 24.2 | 6.8 | 855.4 | 124 | 68.1 | IA,MN |
| | 07080102 | 51.2 | 6.5 | 912.9 | 51 | 70.1 | IA,MN |
| | 07020007 | 16.5 | 6.1 | 778.0 | 69 | 73.2 | MN |
| | 07090006 | 37.8 | 5.9 | 935.2 | 18 | 68.8 | IL,WI |
| | 07060006 | 24.4 | 5.6 | 911.8 | 44 | 59.3 | IA |
| | 07130009 | 20.6 | 5.6 | 1005.0 | 124 | 80.9 | IL |


### 4.3 Future water quality change

Existing studies suggested that CP-El Niño episodes occurred more frequently in a warming climate (Yeh et al., 2009; Yu et
al., 2010). We found that annual loads of TN and TP tend to increase in the Corn Belt region during CP-ENYs in the current
climate. As CP-El Niño frequency increases in the future, TN and TP loads would likely increase over the Corn Belt region
even under the same agricultural conditions, indicating a possible deterioration of water quality in this region when the
climate warms.
The spatial patterns of TN and TP anomalies during CP-ENYs (Fig. S6) also suggested that specific watersheds,
predominantly in southern UMRB and western OTRB, such as Iowa, Illinois, Wisconsin, Indiana, and Kentucky, will likely





experience the most increases of TN and TP loads in the future. Such information is critical to ensure proper decision-
making for watershed protection.

### 4.4 Limitations and future work

The model evaluation suggested that SWAT reasonably captured the hydrological and water quality behaviors in the Corn
Belt. However, this result could be influenced by the uncertainty of model simulations. For example, the limited number of
observation sites might bring uncertainties on the regional scales. The model was assessed by the best available observation
data, and the good agreement between the calculations and observations at 15 sites showed that the model reasonably
captured the changes in the water flow, TN, and TP. However, further assessment of the model is needed when more
observations are available.
The impacts of water temperature change on water quality during different El Niños were not analyzed in the study
because the influences of water temperature and runoff on water quality were coupled in the model. Numerical experiments
considering water temperature changes due to El Niños should be carried out and investigated in future studies.
In addition, as Chen et al. (2021) suggested that the CP-El Niño needed to be further classified into CP-I and CP-II types
due to the differences in sea surface temperature (SST) evolution patterns and climate impacts, the distribution of nutrients in
these two CP-El Niños might also differ and therefore are needed to be further studied in the future.

### 5 Conclusions

The impacts of EP- and CP-El Niños on water quality were investigated by using the SWAT model in the U.S. Corn Belt
region. Calibration and validation results indicated that the simulated streamflow and water quality generally agreed with
observations at most USGS gages. Then, the common features of the annual and seasonal loads of TN and TP for the EP-
and CP-El Niño events in the OTRB and UMRB were analyzed using a composite method based on the simulation results
during 1975–2016.
Annual composite results suggested that TN and TP loads decreased by 13.1% and 14.0% during EP-ENYs, respectively,
whereas they increased by 3.3% and 4.6% during CP-ENYs at the outlet of the OTRB, respectively. TN and TP also showed
a similar pattern at the outlet of the UMRB (18.5% and 19.8% reductions during EP-ENYs, and 5.7% and 4.4% increases
during CP-ENYs, respectively). Furthermore, more sub-basins showed negative annual anomalies of TN and TP during EP-
ENYs with maximum reductions up to −11.7 and −0.9 kg ha$^{-1}$, which were comparable to the normal year mean values (12.7
kg of N ha$^{-1}$ and 1.0 kg of P ha$^{-1}$), respectively, whereas they showed positive anomalies during CP-ENYs.
Seasonal composite results confirmed that water quality anomalies showed opposite patterns during EP-ENYs and CP-
ENYs and the changes in the water quality matched the ENSO phases at the outlet of the OTRB. Maximum reduction or
increase of the nutrients during EP-ENYs or CP-ENYs, respectively, occurred in winter, the peak season of El Niño. At the
outlet of the UMRB (corn-growing region), TN and TP anomalies were also influenced by agriculture activities and became



greater in spring and summer during both EP-ENYs and CP-ENYs, in consistent with the distribution of rainfall changes in
the basin. These results suggested that small changes in climate variables such as precipitation in the growing season could
have greater impacts on water quality in the UMRB during El Niño events.
Our analysis also found that at the outlets of UMRB and OTRB, EP-El Niño had a much greater impact on TN and TP
than CP-El Niño at both annual and seasonal time scales; but CP-El Niño could have comparable impacts on water quality as
EP-El Niño at specific watersheds or hot spots, predominantly in Iowa, Illinois, Minnesota, Wisconsin, and Indiana, which
contributed the greatest amounts of nutrient change to the whole basin loads.
Examination of the climate factors/processes on water quality change indicated that El Niño-induced precipitation and
temperature changes altered runoff and evaporation, and thus TN and TP in both UMRB and OTRB on annual and seasonal
time scales, as well as at the outlet and sub-basin scales. It is also found that nutrient levels were largely determined by
precipitation through runoff during both EP- and CP-ENYs, especially at the outlet, as the precipitation was a major source
of runoff, and nitrogen and phosphorus components were transported by runoff. At the sub-basin scale, water quality was
affected by the combination of precipitation and agricultural activities, especially in the UMRB during the growing season.
In the future when the climate continues to warm, the CP-El Niño episode is projected to occur more frequently, TN and
TP loads might increase in the Corn Belt region even under the same agricultural conditions, while water quality would
generally get better in EP-El Niño years. The findings from this study may help ensure proper decision-making for
watershed protection and possible ways to address anticipated water quality change associated with El Niño events.

*Data availability.* Available upon request.
*Author Contributions.* W.L. and P.C. designed the research; P.C. and K.H. performed the data collection and model
calculation; W.L. and P.C. contributed to the interpretation of the results; P.C. and W.L. wrote the manuscript.
*Competing Interests.* The authors declare no competing interests.
*Disclaimer.* Publisher's note: Copernicus Publications remains neutral with regard to jurisdictional claims in published maps
and institutional affiliations.
*Acknowledgments.* We thank Dr. Yongping Yuan for the suggestions on simulations of water quality in the OTRB and
UMRB. This research was supported by the Natural Science Foundation for Young Scientists of Shanxi Province, China (No.
202103021223106 and 20210302124458), the National Key Research and Development Program, China (No.
2018YFC0406406), the China Scholarship Council (File No. 201806935023), the Institute-Local Cooperation Project of the
Chinese Academy of Engineering (No. 2020SX8), the Government Financial Grants Project, China (No. ZNGZ2015-036),
the Key Research and Development Program (Social Development Field) of Shanxi Province, China (No. 201903D321052),
and the Natural Science Foundation of Shanxi Province, China (No. 201901D111059 and 201901D111060).



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
