# Peer review of "Impacts of different types of El Niño events on water quality over the"

_Hydrology and Earth System Sciences, 2022_

## Author Response (AR1)

Dear Dr. Stacey Archfield,

We appreciate you and anonymous referees for the helpful and inspiring comments and have implemented all the suggestions in the revised manuscript. The following texts are our point-to-point responses, all comments are in black, and the replies are in red.

**Responses to editor**

**- For Reviewer #2, Comment #3 Response: Please add more detail as to how the significance level is determined from the 500 simulation runs. Is this simply a calculation of the probability of random occurrence obtained from the simulation versus what was observed? Is so, please add this detail so that it is clear to the reader. If I have this incorrect, please add the correct detail.**

**Reply:** We have added detailed information on calculating significance levels as follows:

Statistical significance of precipitation, temperature, runoff, evaporation, TN, or TP anomalies in El Niño years was tested by the Monte Carlo method (Mo, 2010). The underlying concept of the method is to use randomness to solve problems that might be deterministic in principle (Wilks, 1995). Taking TN as an example, in order to test whether TN anomalies in EP-El Niño years were significantly different from those in normal years, we first composited (i.e., averaged) TN anomalies for the nine EP-El Niño years (1976–1977, 1979–1980, 1982–1983, 1986–1987, 1987–1988, 1991–1992, 1997–1998, 2006–2007, and 2015–2016). The composite analysis is a useful technique to determine some of the basic structural characteristics of a climatological phenomenon, such as El Niño which occur over time. We then randomly selected nine years out of 1975-2016 (i.e., keeping the same number of years as the EP-El Niño years) and averaged/composited TN anomalies for the nine

randomized years as the first sample. The process was repeated 500 times. These composite samples were used to generate a distribution corresponding to the null hypothesis, against which we could evaluate whether TN anomalies during EP-El Niño were significantly different from those in normal years at a 95% confidence level. Similarly, significance levels of the composite results of precipitation, temperature, runoff, evaporation, and TP anomalies in EP-El Niño years could be determined. Such a method has been widely used in climate-related studies (Laken and Čalogović, 2013; Mo, 2010; Sanchez and Karnauskas, 2021) due to its robustness. Please find the added information on Pg. 5-6, line 97-109.

**Responses to Referees 1**

The authors applied a standard SWAT model in two large basins to undertstand the impacts of Eastern Pacific (EP) and Central Pacific (CP) El Niños on water quality. They found contrasting water quality effects due to differences in precipitation and air temp annormalies between the two EI Niños.

(1) The authors suggest that impacts of extreme climate on the load of N and P to the rivers are dominated by variability of precip and consequently runoff. They discussed very little of impacts of hydrological change on biogeochemical cycles - the basiss of water quality change. For example, how the change in temp affects N denitrification and carbon decomposition and N leaching processes, in addition to water quantity through ET? Change in climate and hydrology is not only affecting total nutrient load but also the concentration of flow chemistry. More discussion in this aspect will provide more insights on the impacts of extreme climate change.

**Reply:** We agree with the reviewer that temperature could also impact nitrogen and phosphorus components, such as nitrate, organic nitrogen, soluble phosphorus, mineral phosphorus, and organic phosphorus besides total nutrient loads during El Niño events. According to the comment, we have carried out new analyses and the results showed that compared to precipitation, temperature plays a secondary role in altering nutrient levels through biogeochemical processes. We have added a new Section 4.3 about biogeochemical process variations due to temperature change as follows.

4.3 Biogeochemical process variations due to temperature change

The effect of temperature on water quality through affecting evaporation and runoff has been analyzed in Section 3.2.2. In fact, the temperature can also affect water quality through some biochemical processes of nutrients (Neitsch et al., 2011). In order to analyze the biogeochemical process variations due to temperature changes during EP- and CP-El Niños, new analyses on nitrogen and phosphorus components, such as nitrate, organic nitrogen, soluble phosphorus, mineral phosphorus, and organic phosphorus had been carried out. Results from the analyses demonstrated that compared to precipitation, temperature plays a secondary role in altering nutrient levels through biogeochemical processes. Taking nitrate as an example, we showed the composite results of annual and seasonal nitrate anomalies (Figs. S6 and S7), respectively, during EP- and CP-El Niños. Figures S6 and S7 indicated that the pattern of nitrate was more similar to that of precipitation (Figs. 6a, 6c, and 7a-7h) but different from that of temperature (Figs. 8a, 8c, and 9a-9h) in the Corn Belt region during El Niños. This could be further confirmed by the pattern correlation results. The correlation coefficients of annual nitrate and precipitation were 0.47, 0.36, 0.22, and 0.39, respectively, at OTRB and UMRB during EP- and CP-El Niños. The correlation coefficients between nitrate and temperature were relatively small (the coefficients

were -0.15, 0.08, 0.30, and -0.31, respectively). The coefficient values altered between positive and negative at the two basins during EP- and CP-El Niños. The inconsistent relationships between nitrate and temperature were mainly because the nitrate content could vary through nitrification, mineralization, denitrification, and plant uptake processes (Neitsch et al., 2011). When the temperature rises, the former two processes increase nitrate content, but the latter two decrease nitrate content. Thus, the final sign of the correlation coefficient between nitrate and temperature really depends on the dominant processes. Similar results were also found at seasonal scales (not shown). These results indicated that nitrate variations were dominated by precipitation variations in the two basins during EP- and CP-El Niños, instead of temperature impacts on the biogeochemical processes. Similar results were also found for other nutrient components, such as organic nitrogen, soluble phosphorus, mineral phosphorus, and organic phosphorus (not shown). (Pg. 21-22, line 413-434)

Thank you for the suggestion.

[Figure]

**Figure S6**. Composite results of annual nitrate (NO₃) anomalies (unit: kg ha$^{-1}$) in EP-El Niño years (a) and in CP-El Niño years (b) during the period of 1975–2016. Stippling denotes anomalies significantly different from zero at the 95% confidence level based on the Monte Carlo test.

[Figure]

**Figure S7**. Seasonal composite of nitrate (NO$_3$) anomalies (unit: kg ha$^{-1}$) in summer (JJA) (a), autumn (SON) (b), winter (DJF) (c), and spring (MAM) (d) in EP-El Niño years during the period of 1975–2016; (e-h) are the same as (a-d) but for CP-El Niño years. Stippling denotes anomalies significantly different from zero at the 95% confidence level based on the Monte Carlo test.

**Responses to Referees 2**
* * *
**The authors investigated different impacts of two El Niño events on water quality over the Corn Belt region of US. The authors find that different El Niño events have different impact on TN and TP levels in the water on both annual and seasonal scales and these impacts are mainly driven by the changes of precipitation, as well as evaporation to a lesser extent. The manuscript is well written. The method of this study is solid and the results are well presented, providing new insights to the community. However, this paper needs some revisions before the acceptance for publication.**

**(1) The Corn Belt region is agricultural important. However, this is not clearly seen in the introduction section (Line 33-39). The authors need to added some sentences to describe why Corn Belt region needs your attention or why the water quality in this region is important, e.g., agriculture production/corn production, the fraction compared with the whole US. Besides, will a higher level of TN and TP in streamflow benefit agriculture or damage agriculture? These background information are missing, but imperative to the readers to highlight the importance of your study.**

**Reply:** We have added background information on the Corn Belt region in the revised manuscript. Please also see below.

The Corn Belt is a very important area of the agricultural activity of the country, as 75% of the corn and 60% of the soybean produced in the U.S. are grown in the region (Thaler et al., 2021). The region's agricultural activities such as fertilizers contribute to the increase of nitrogen and phosphorus levels, which are responsible for the Gulf of Mexico hypoxic zone (Panagopoulos et al., 2014, 2015; Rabalais et al., 2007). The required nutrient reduction of the Corn Belt to decrease hypoxia is the highest among

all regions in the Mississippi River Basin (Mississippi River/Gulf of Mexico Watershed Nutrient Task Force, 2011). Hence, water quality changes in the Corn Belt region have been receiving considerable attention. (Pg. 2, line 32-38)

**(2) Section 3.1.1 (1), a significance test is missing. Besides, why the results are shown in a table while for the results on seasonal scale (3.1.2 (1)) are displayed in bar plot? Maybe the authors should keep them consistent, all showing in bar plot. For the bar plot, an error bar should be added to show the spread.**

**Reply:** We have removed Table 4 and re-plotted it as Fig. 2 in the revised manuscript. Figure 2 shows the detailed statistical information including the mean, median, $25^{th}$ and $75^{th}$ percentile, and the $10^{th}$ and $90^{th}$ percentile, of TN and TP at the outlets of the OTRB and UMRB during EP- and CP-El Niño years. To be consistent with the nutrients on the annual scale (Fig. 2), we also re-plotted Fig. 4 to replace the previous Fig. 3 in Section 3.1.2 on seasonal scales following the comment.

[Figure]

**Figure 2**. Box plots of annual (a) TN and (b) TP anomalies (unit: $10^3$ tons) at the outlets of the OTRB and UMRB during EP-El Niño years and CP-El Niño years, respectively. The green plus (+), red solid horizontal line, box, and whisker ends indicate the mean, median, $25^{th}$ and $75^{th}$ percentile, and the $10^{th}$ and $90^{th}$ percentile, respectively. The data points outside the ranges are shown in hollow dots.

[Figure]

**Figure 4**. Same as Fig. 2 but for seasonal scales, i.e., summer (June-August, JJA), autumn (September-November, SON), winter (December of the current year and January and February of the following year, DJF), and spring (March-May, MAM).

**(3) How the Monte Carlo test is performed in your study? This is also missing in the methods section.**

**Reply:** Monte Carlo tests were performed following Mo (2010). We have added the details of the Monte Carlo test in Section 2.1 (Data). Please see below:

Statistical significance of precipitation, temperature, runoff, evaporation, TN, or TP anomalies in El Niño years was tested by the Monte Carlo method (Mo, 2010). The underlying concept of the method is to use randomness to solve problems that might be deterministic in principle (Wilks, 1995). Taking TN as an example, in order to test whether TN anomalies in EP-El Niño years were significantly different from those in normal years, we first composited (i.e., averaged) TN anomalies for the nine EP-El Niño years (1976–1977, 1979–1980, 1982–1983, 1986–1987, 1987–1988, 1991–1992, 1997–1998, 2006–2007, and 2015–2016). The composite analysis is a useful technique to determine some of the basic structural characteristics of a climatological phenomenon, such as El Niño which occur over time. We then randomly selected nine years out of 1975-2016 (i.e., keeping the same number of years as the EP-El Niño years) and averaged/composited TN anomalies for the nine randomized years as the first sample. The process was repeated 500 times. These composite samples were used to generate a distribution corresponding to the null hypothesis, against which we could evaluate whether TN anomalies during EP-El Niño were significantly different from those in normal years at a 95% confidence level. Similarly, significance levels of the composite results of precipitation, temperature, runoff, evaporation, and TP anomalies in EP-El Niño years could be determined. Such a method has been widely used in climate-related studies (Laken and Čalogović, 2013; Mo, 2010; Sanchez and Karnauskas, 2021) due to its robustness. Please find the added information on Pg. 5-6, line 97-109.

**(4) More description of the model is needed. For example, what is the forcing data of the model? Does the forcing data include the two El Niño events? The resolution of the model?**

**Reply:** The information on forcing data could be found in Section 2.1 (Data). Please see below:

The weather data (i.e., forcing data of the SWAT model), including precipitation and temperature, were obtained from 2,242 National Weather Service (NWS) stations in the study area. The forcing data included CP and EP-Niño events. Specifically, nine EP-El Niño events (1976–1977, 1979–1980, 1982–1983, 1986–1987, 1987–1988, 1991–1992, 1997–1998, 2006–2007, and 2015–2016) and six CP-El Niño events (1977–1978, 1990–1991, 1994–1995, 2002–2003, 2004–2005, and 2009–2010) occurred during the study period (1975–2016). (Pg. 3, line 81-85)

According to the comment, we also added more descriptions of the model, such as the spatial and temporal resolutions of the model in Section 2.2 (SWAT model description). Please see below:

In the SWAT model, a basin is partitioned into sub-basins, which are further divided into hydrological response units (HRUs). Runoff, sediment, and nutrient loads are simulated for each HRU and then aggregated for sub-basins. Thus, the spatial resolution of the model is measured by the number of HRUs and sub-basins. In total, the OTRB and UMRB included 152 and 131 sub-basins, respectively, and a total of 20,157 and 20,581 HRUs in the OTRB and UMRB. The model was calculated on a daily time scale and the results were analyzed on a monthly time scale. (Pg. 6, line 114-123)

**(5) On seasonal scales, the authors find that the changes of nutrients level are stronger in spring and summer. However, El Niño is usually strongest during winter. Is there any explanation for this delay?**

**Reply:** The explanation for stronger signals in spring and summer could be found in Section 4.1 as follows.

On seasonal scales, changes in nutrients' magnitudes were stronger in spring and summer, especially in UMRB. The heavy loading of nutrients was related to the agriculture activities during the growth period of crops in the Corn Belt. The major crops here are corn and soybean, which are often planted and fertilized in May and harvested in October (Chiang et al., 2014). Hence, the higher nutrient levels were likely associated with the removal of fertilizers from the soil during spring and summer. (Pg. 20, line 383-387)

**(6) Line 327-328, in CP-ENYs, temperature decreased insignificantly, but evaporation increased significantly. Is there any explanation for this phenomenon, as by intuition, evaporation should decrease as temperature decreases.**

**Reply:** The explanation for different patterns of evaporation and temperature could be found in Section 3.2.2 as follows:

Figures 8 showed that changes in evaporation did not share the same pattern with temperature change on the annual time scale. This might be due to the fact that temperature directly affected potential evapotranspiration (Neitsch et al., 2011), the ability of the atmosphere to remove water from the surface through both evaporation and transpiration; but the actual evaporation/evapotranspiration was also related to other variables such as the amount of water available for evaporation besides temperature. (Pg. 17, line 328-333)

**(7) The authors identify that precipitation is the most crucial factor that influencing TP and TN concentration by controlling runoff. Does irrigation have an impact on runoff or nutrient level?**

**Reply:** We agree with the reviewer that irrigation could impact runoff and thus nutrient levels; however, it is hard to quantify the exact effect of irrigation due to lack of the irrigation data over the Corn Belt. Existing documents suggest that vast acreages of corn and soybeans are watered by center pivot irrigation in the Corn Belt region, which uses an apparatus that sprays water across a field with a 75–90% efficiency, thus irrigation water mostly infiltrates into the soil (Grassini et al., 2011; 2014; Green et al., 2018). Precipitation likely plays a dominant role in runoff, we thus focus on the impact of precipitation on runoff and water quality in the study. We plan to test the results once detailed irrigation data are available. We have added the sentences in Section 4.5 (limitations and future work) following the comment. (Pg. 22-23, line 452-457)

**References**

Chiang, L. C., Yuan, Y., Mehaffey, M., Jackson, M., and Chaubey, I.: Assessing SWAT's performance in the Kaskaskia River watershed as influenced by the number of calibration stations used, 28, 676–687, https://doi.org/10.1002/hyp.9589, 2014.

Grassini, P., Yang, H., Irmak, S., Thorburn, J., Burr, C., and Cassman, K. G.: High-yield irrigated maize in the Western U.S. Corn Belt: II. Irrigation management and crop water productivity, F. Crop. Res., 120, 133–141, https://doi.org/10.1016/j.fcr.2010.09.013, 2011.

Grassini, P., Torrion, J. A., Cassman, K. G., Yang, H. S., and Specht, J. E.: Drivers of spatial and temporal variation in soybean yield and irrigation requirements in the

western US Corn Belt, 163, 32–46, https://doi.org/10.1016/j.fcr.2014.04.005, 2014.

Green, T. R., Kipka, H., David, O., and McMaster, G. S.: Where is the USA Corn Belt, and how is it changing?, Sci. Total Environ., 618, 1613–1618, https://doi.org/10.1016/j.scitotenv.2017.09.325, 2018.

Laken, B. A. and Čalogović, J.: Composite analysis with monte carlo methods: An example with cosmic rays and clouds, 3, https://doi.org/10.1051/swsc/2013051, 2013.

Mississippi River/Gulf of Mexico Watershed Nutrient Task Force: Gulf Hypoxia: Action Plan 2008 for Reducing, Mitigating and Controlling Hypoxia in the Northern Gulf of Mexico and Improving Water Quality in the Mississippi River Basin, in Hypoxia in the Northern Gulf of Mexico, pp. 265–305., 2011.

Mo, K. C.: Interdecadal modulation of the impact of ENSO on precipitation and temperature over the United States, J. Clim., 23(13), 3639–3656, doi:10.1175/2010JCLI3553.1, 2010.

Neitsch, S. L., Arnold, J. G., Kiniry, J. R. and Williams, J. R.: Soil and Water Assessment Tool Theoretical Documentation Version 2009, Available online: http://hdl.handle.net/1969.1/128050., 2011.

Panagopoulos, Y., Gassman, P. W., Arritt, R. W., Herzmann, D. E., Campbell, T. D., Jha, M. K., Kling, C. L., Srinivasan, R., White, M. and Arnold, J. G.: Surface water quality and cropping systems sustainability under a changing climate in the Upper Mississippi River Basin, J. Soil Water Conserv., 69(6), 483–494, doi:10.2489/jswc.69.6.483, 2014.

Panagopoulos, Y., Gassman, P. W., Jha, M. K., Kling, C. L., Campbell, T., Srinivasan, R., White, M. and Arnold, J. G.: A refined regional modeling approach for the

Corn Belt - Experiences and recommendations for large-scale integrated modeling, J. Hydrol., 524, 348–366, doi:10.1016/j.jhydrol.2015.02.039, 2015.

Rabalais, N. N., Turner, R. E., Sen Gupta, B. K., Boesch, D. F., Chapman, P. and Murrell, M. C.: Hypoxia in the northern Gulf of Mexico: Does the science support the plan to reduce, mitigate, and control hypoxia?, Estuaries and Coasts, 30(5), 753–772, doi:10.1007/BF02841332, 2007.

Sanchez, S. C. and Karnauskas, K. B.: Diversity in the Persistence of El Niño Events Over the Last Millennium, Geophys. Res. Lett., 48, e2021GL093698, https://doi.org/10.1029/2021GL093698, 2021.

Thaler, E. A., Larsen, I. J., and Yu, Q.: The extent of soil loss across the US Corn Belt, 118, 1–8, https://doi.org/10.1073/pnas.1922375118, 2021.

Wilks, D. S.: Statistical Methods in the Atmospheric Sciences: An Introduction. Academic press, 1995.